# Parent-of-origin effects propagate through networks to shape metabolic traits

Juan F Macias-Velasco[1], Celine L St Pierre[1], Jessica P Wayhart[1], Li Yin[2], Larry Spears[2], Mario A Miranda[1], Caryn Carson[1], Katsuhiko Funai[3], James M Cheverud[4], Clay F Semenkovich[2], Heather A Lawson[1]*

[1]Department of Genetics, Washington University School of Medicine, Saint Louis, United States; [2]Department of Medicine, Washington University School of Medicine, Saint Louis, United States; [3]Diabetes and Metabolism Research Center, University of Utah, Salt Lake City, United States; [4]Department of Biology, Loyola University, Chicago, United States

**Abstract** Parent-of-origin effects are unexpectedly common in complex traits, including metabolic and neurological traits. Parent-of-origin effects can be modified by the environment, but the architecture of these gene-by-environmental effects on phenotypes remains to be unraveled. Previously, quantitative trait loci (QTL) showing context-specific parent-of-origin effects on metabolic traits were mapped in the $F_{16}$ generation of an advanced intercross between LG/J and SM/J inbred mice. However, these QTL were not enriched for known imprinted genes, suggesting another mechanism is needed to explain these parent-of-origin effects phenomena. We propose that non-imprinted genes can generate complex parent-of-origin effects on metabolic traits through interactions with imprinted genes. Here, we employ data from mouse populations at different levels of intercrossing ($F_0$, $F_1$, $F_2$, $F_{16}$) of the LG/J and SM/J inbred mouse lines to test this hypothesis. Using multiple populations and incorporating genetic, genomic, and physiological data, we leverage orthogonal evidence to identify networks of genes through which parent-of-origin effects propagate. We identify a network comprised of three imprinted and six non-imprinted genes that show parent-of-origin effects. This epistatic network forms a nutritional responsive pathway and the genes comprising it jointly serve cellular functions associated with growth. We focus on two genes, *Nnat* and *F2r*, whose interaction associates with serum glucose levels across generations in high-fat-fed females. Single-cell RNAseq reveals that *Nnat* expression increases and *F2r* expression decreases in pre-adipocytes along an adipogenic trajectory, a result that is consistent with our observations in bulk white adipose tissue.

*For correspondence: lawson@wustl.edu

Competing interest: The authors declare that no competing interests exist.

## Editor's evaluation

We all learned simple Mendelian Punnett Squares in Junior High or earlier when studying simple Mendelian traits. But we also all know that the world is so much richer and more complex than this. The current article explores some of that complexity, opening rich insights into intergenerational effects, offering the opportunity for mathematical thinking and further hypothesis testing, and opening up exciting new hypotheses to test. As Professor Stephen Stearns wrote, "Many of us do not do science only, or even primarily, to achieve practical results. We do it because we are fascinated with neat ideas. Evolutionary medicine is full of them, including parent-of-origin pattern." Let us enjoy the wonder.

## Introduction

Parent-of-origin effects, where the phenotypic effect of an allele depends on whether the allele is inherited maternally or paternally, are epigenetic phenomena associated with a wide range of complex traits and diseases (*Lawson et al., 2013*). Thus, the functional impact of a specific genetic variant can depend on its parental origin. The best characterized parent-of-origin effect is genomic imprinting, an epigenetic process in which either the maternally or paternally inherited allele is silenced, typically through DNA methylation. In humans, there are 107 verified imprinted genes and in mice there are 124, of which ~ 70% overlap (*Jirtle, 2012*). Despite the rarity of imprinted genes, parent-of-origin effects on complex traits and diseases are relatively common, suggesting that canonical imprinting mechanisms are not sufficient to account for these phenomena (*Mozaffari et al., 2019*; *Zeng et al., 2019*). With so few imprinted genes, what mechanisms underlie these parent-of-origin effects? We hypothesize that a small number of imprinted genes can generate a large number of parent-of-origin effects through interactions with non-imprinted genes.

In this study, we use four populations at different levels of intercrossing of the LG/J and SM/J inbred mouse lines to test the hypothesis that non-imprinted genes can contribute to parent-of-origin effects on metabolic phenotypes through epistatic interactions with imprinted genes. Multiple populations ($F_0$, $F_1$, $F_2$, $F_{16}$) allow us to refine our search space and provide orthogonal evidence supporting putative networks of interacting genes. Metabolic traits were previously mapped in a $F_{16}$ generation of an advanced intercross between LG/J and SM/J (*Cheverud et al., 2011*; *Lawson et al., 2010*; *Lawson et al., 2011a*; *Lawson et al., 2011b*). We generated visceral white adipose tissue gene expression profiles from 20 week-old $F_1$ animals in order to match the age of the $F_{16}$ LG/J x SM/J advanced intercross population. $F_1$ reciprocal cross (LxS and SxL) mice were subjected to the same high and low-fat diets and phenotyping protocols as the previously-studied $F_{16}$ mice to keep environmental contexts consistent. We identified genes showing parent-of-origin-dependent allele-specific expression (ASE), characterized interactions among these genes and biallelic genes that are differentially expressed by reciprocal cross (DE), and correlated interacting ASE and DE gene pairs with metabolic phenotypes in the $F_1$ population. Pairs that significantly associated with phenotypic variation were tested for epistasis on correlated traits in the $F_{16}$ population.

We identified an epistatic network that forms a nutritional environment responsive pathway mediated through calcium signaling. This network contributes to metabolic variation by balancing proliferation, differentiation, and apoptosis in adipocytes. The genes comprising this network jointly serve functions associated with growth in multiple tissues, which is consistent with the evolutionary hypothesis that sexual conflict underlies some parent-of-origin effects (*Mochizuki et al., 1996*). We focus on two key interacting genes: *Nnat* (neuronatin), a canonically imprinted gene, and *F2r* (coagulation factor II receptor), a biallelic gene showing significant DE by cross in $F_1$ high-fat-fed female animals. Co-expression of these two genes associates with variation in basal glucose levels, and this association persists across generations. Further, single-cell RNAseq reveals that *Nnat* expression increases and *F2r* expression decreases in pre-adipocytes along an adipogenic trajectory, a pattern consistent both with their expression in bulk white adipose tissue and with their respective roles in adipogenesis. Our results demonstrate that incorporating orthogonal lines of evidence including genotype, allele specific expression, total gene expression, single-cell expression, and phenotype from different populations varying in their degree of intercrossing is a powerful way to identify putative mechanisms and test hypotheses underlying parent-of-origin effects on phenotype.

## Results

### Non-imprinted genes interact with imprinted genes and effect metabolic phenotypes

We test the hypothesis that non-imprinted genes can mediate complex parent-of-origin effects on phenotypes through genetic interactions with imprinted genes using a $F_1$ reciprocal cross model of the LG/J and SM/J inbred mice (LxS and SxL). In this model the effects of parental origin on an allele can be tested directly and isolated from sequence dependent *cis*-regulatory differences. We validated our findings in LG/J and SM/J parentals ($F_0$) as well as in $F_2$ and $F_{16}$ intercrosses of LGxSM (*Figure 1*). The parental $F_0$ animals serve to anchor variation in allele-specific expression that is a function of allelic identity (L or S). Incorporating the $F_2$ and $F_{16}$ populations into our validations ensures

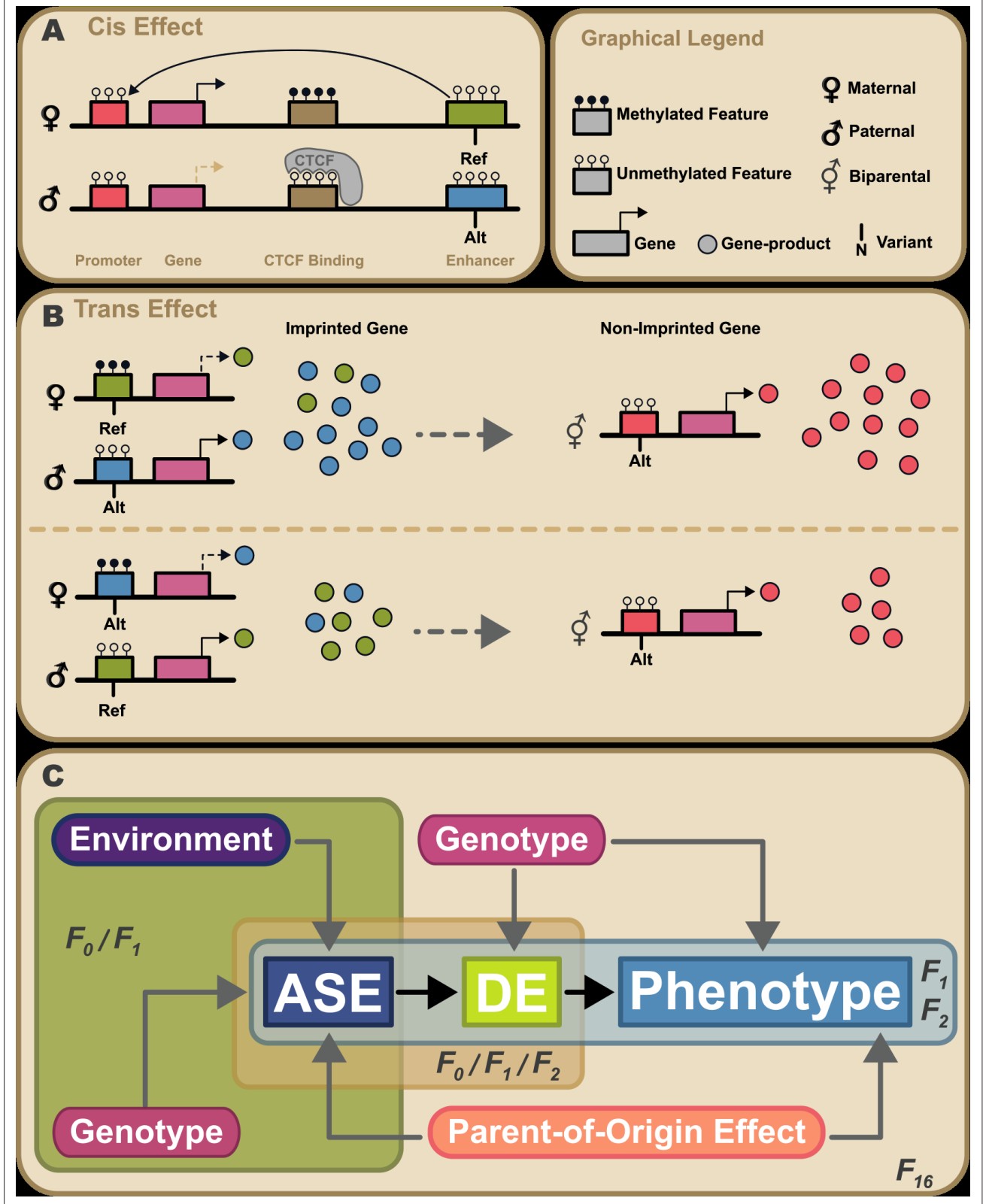

**Figure 1.** Proposed model for propagation of parent-of-origin effects through gene-gene interactions. Parent-of-origin effects should be partitioned into *cis* mechanisms and *trans* mechanisms (**A**) An example of a *cis* parent-of-origin effect is a system with three regulatory elements: promoter, insulator, and enhancer. Activation of transcription requires the enhancer to act upon the promoter. Enhancer activity is blocked by the insulator when it has been bound by CTCF. CTCF cannot be bound when methylated. In this system, the insulator is selectively methylated when inherited maternally,

*Figure 1 continued on next page*

*Figure 1 continued*

so methylation of the maternally inherited insulator blocks CTCF binding, allowing the enhancer to activate transcription. Because the paternally inherited insulator is not methylated, it is bound by CTCF which blocks enhancer activity, silencing transcription. This canonical genomic imprinting mechanism interacts with genetic variation in the three regulatory features. For example, if one allele produces stronger enhancer activity (Alt) than the other, individuals inheriting the Alt allele maternally would have elevated expression compared to those that inherit the same allele paternally. These *cis* genetic effects do not occur in isolation. Due to the highly interconnected nature of biological systems, there are downstream effects. We refer to these as *trans* parent-of-origin effects. (**B**) An example of a *trans* parent-of-origin effect is a system with two genes each having its own promoter. The first gene is canonically imprinted, and the activity of the gene promoter is blocked by DNA methylation. The imprinted gene's promoter is methylated when inherited maternally. Consequently, the paternally inherited allele is almost exclusively expressed. As before, when genetic variation in a regulatory feature interacts with these epigenetic mechanisms, we see parent-of-origin effects on expression of the imprinted gene. In this example, the imprinted gene regulates expression of a non-imprinted gene. Despite the non-imprinted gene being agnostic to parental origin, its expression nonetheless depends on the parental origin of alleles at the imprinted locus. (**C**) Summary of our experimental design. Expression patterns of genes showing allele-specific expression (ASE) such as imprinted genes are shaped by parental genotypes and environment (e.g. nutrition). Downstream gene expression is a function of their genotype and the expression of upstream ASE genes. Altered parent-of-origin dependent total gene expression of ASE genes leads to differential expression of downstream genes varying only in allelic parent-of-origin (DE). Phenotype is most directly affected by expression of DE genes. Variation in DE gene expression leads to corresponding variation in phenotype. Mouse populations used to probe parts of this model are labeled $F_0$ (inbred lines), $F_1$ (reciprocal cross of inbred lines), $F_2$ (intercross of $F_1$ mice), and $F_{16}$ (advanced intercross of inbred lines).

that the interactions we observe are not solely a function of linkage in the $F_1$ animals. We generated mRNA expression profiles in white adipose tissue from 20-week-old $F_1$ reciprocal cross animals. These animals were subjected to the same high and low-fat diets and phenotyping protocols as the previously studied $F_{16}$ animals (*Cheverud et al., 2011*; *Lawson et al., 2011a, Lawson et al., 2010*; *Carson et al., 2020*; *Miranda et al., 2020*). We identified two classes of genes: (1) imprinted genes and (2) non-imprinted genes with parent-of-origin effects on total expression.

To test our model, we identified genes showing parent-of-origin dependent allele specific expression (ASE). We identified 23 genes showing significant ASE (*Figure 2A*; *Supplementary file 1*). Of these 23 genes, 17 are canonically imprinted genes, two are not reported as imprinted genes but are located in known imprinted domains, and four are novel. Next, we identified genes showing differential total expression between individuals varying only in allelic parent-of-origin (DE between reciprocal crosses, SxL vs LxS). We identified 33 genes that are significantly DE in at least one sex or dietary context (*Figure 2A*; *Supplementary file 2*). A larger set of genes show signatures of parent-of-origin effects at the total gene expression level, but do not meet the statistical rigor demanded by the massive multiple tests burden incurred by a genome-wide scan accounting for sex, diet, and parent-of-origin (see Materials and methods).

To identify interactions between gene sets, we constructed a network comprised of genes that could initiate a parent-of-origin effect on phenotype (ASE) and genes that may mediate the effect onto phenotype (DE). Interacting gene pairs were predicted by modeling the expression of biallelic genes that are significantly DE by reciprocal cross as a function of the expression of genes showing significant parent-of-origin-dependent ASE, their allelic bias ($L_{bias}$), diet, sex, and the diet-by-sex interaction. Genes showing parent-of-origin effects form a highly interconnected network comprised of 52 genes forming 217 gene pairs (*Figure 2B*)(*Supplementary file 3*). Most of these interactions are *trans*-chromosomal. We identified two genes that could serve as initiation points of propagating parent-of-origin effects through this network. These two genes, *Nnat* (neuronatin) and *Cdkn1c* (cyclin dependent kinase inhibitor 1 C), are both canonically imprinted and differentially expressed by reciprocal cross (*Supplementary file 1*).

Functional over representation analysis was performed and seven terms were significantly over-represented at an FDR ≤ 0.05 (*Figure 2C*; *Zhang et al., 2005*). Enriched terms suggest this network plays a role in signaling and genetic imprinting (*Supplementary file 4*). In order to identify which phenotypes might be affected by genes in this network, gene expression was correlated with metabolic phenotypes collected for the $F_1$ animals (*Figure 2D*). Seventy-four ASE/DE/phenotype sets were identified as candidates for subsequent testing (*Supplementary file 5*).

## Epistasis in an $F_{16}$ advanced intercross identifies a diet-responsive network affecting adipogenesis

To validate the interactions we identified in $F_1$ animals, we tested for imprinting-by-imprinting epistasis in an $F_{16}$ population. Imprinting-by-imprinting epistasis occurs when the parent-of-origin effect at

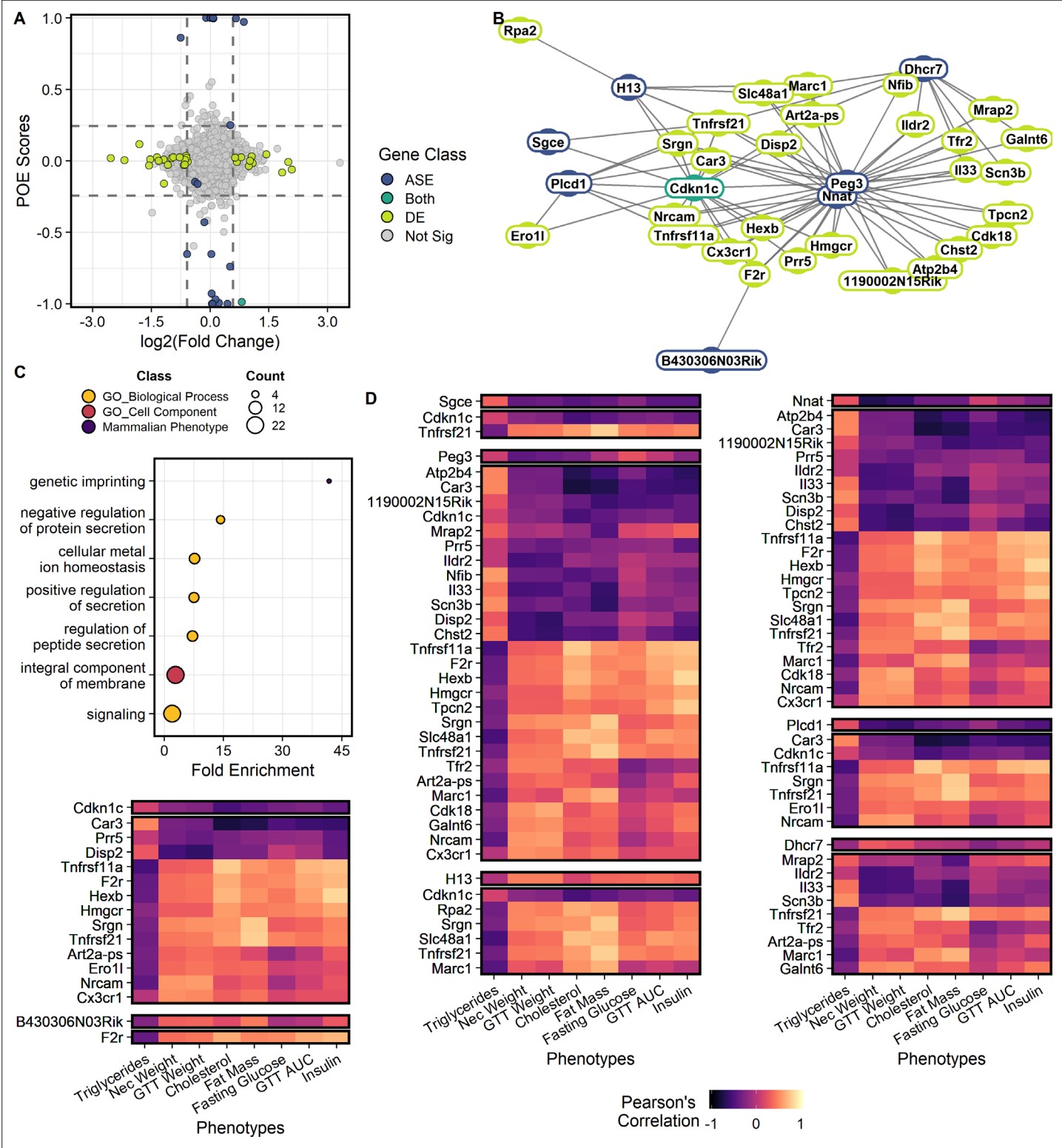

**Figure 2.** Genes showing parent-of-origin effects at the allele specific and/or total expression levels covary with each other and with metabolic traits. (**A**) Mean parent-of-origin effect score across contexts. Effect size of ASE is calculated as the mean allelic bias (L / L + S) of SxL animals minus LxS animals. Effect size of DE is measured by log$_2$(Fold Change) between LxS and SxL crosses. The single context with largest magnitude fold change is plotted for each gene. Dashed lines represent minimum acceptable effect size cut-offs within a context. Genes showing significant ASE and sufficiently large parent-of-origin effect score are shown in blue. Genes showing significant DE and sufficiently large fold change in some sex or dietary context are shown in lime. Genes showing both ASE and DE are shown in teal. Genes not meeting cut-offs are shown in gray. The two genes showing significant

*Figure 2 continued on next page*

*Figure 2 continued*

ASE but falling short of parent-of-origin effect score requirements are a case of context dependent bipolar parent-of-origin effect scores (i.e. paternally expressed in one context and maternally expressed in its opposite). (**B**) Parent-of-origin effect network constructed from ASE and DE gene pairs. (**C**) Significantly overrepresented ontologies after multiple tests correction in the parent-of-origin effect network. Terms are color coded by ontology domain. GO biological process (yellow), GO cellular component (orange), and Mammalian phenotype (purple). Circle size denotes the number of genes with each term. (**D**) Correlation of parent-of-origin effect network genes with metabolic traits. Only genes and phenotypes with at least one significant correlation after multiple test corrections are shown. The heatmap is broken up into subnetworks with the ASE gene as the first separated row followed by associated DE genes in subsequent rows. Columns correspond to metabolic traits. Coloration of each cell denotes the Pearson's correlation coefficient value.

The online version of this article includes the following figure supplement(s) for figure 2:

**Figure supplement 1.** RNAseq libraries are sufficiently complex to detect allele specific expression.

**Figure supplement 2.** Number of reads mapped to LG/J x SM/J pseudo-genome.

**Figure supplement 3.** Stable null permutation plots for allele-specific expression.

**Figure supplement 4.** Multiple tests correction of ASE detection.

**Figure supplement 5.** Volcano plots of parent-of-origin dependent allele-specific expression.

**Figure supplement 6.** Multiple tests correction of DE detection.

**Figure supplement 7.** Volcano plots of differentially expressed genes.

**Figure supplement 8.** Stable null permutation plots for network pairs.

**Figure supplement 9.** Multiple tests correction of pairwise network construction.

**Figure supplement 10.** Volcano plots of network construction.

**Figure supplement 11.** Example transformation of $F_1$ phenotypes.

**Figure supplement 12.** Multiple tests correction of phenotype correlations.

**Figure supplement 13.** Volcano plots of phenotypes correlated with POE net gene expression.

a locus is dependent on the parent-of-origin of alleles at another locus. This allowed us to determine if the effect of parent-of-origin at DE genotype on phenotype is dependent upon the parent-of-origin at ASE genotype. This orthogonal approach allows us to connect genotype at these loci to phenotype as predicted in the $F_1$ candidates. Nine epistatic interactions replicated in the $F_{16}$ population (n = 1002 animals, FDR ≤ 0.1; *Figure 3A*; *Supplementary file 6*). These interactions were comprised of three ASE genes showing parent-of-origin (*Cdknlc*, *Nnat*, *Plcd1*), six genes that are DE by cross (*Car3*, *F2r*, *Hexb*, *Hmger*, *Srgn*, *Tnfrsf11a*) and four phenotypes (basal glucose level, AUC calculated from a glucose tolerance test, serum cholesterol, necropsy weight). Together, these nine genes form a putative diet-responsive network affecting adipogenesis (*Figure 3B*).

The network can be broken down into signal potentiation, transduction, and response. *Nnat* (neuronatin) and *Hexb* (beta-hexosaminidase subunit beta) fall into the potentiation group. These genes play a role in managing the availability and accumulation of calcium necessary for signal transduction. *Nnat* is a paternally expressed canonically imprinted gene which encodes a proteolipid protein that localizes to the ER (*Li et al., 2010*). *Nnat* is diet-responsive and its overexpression in 3T3L1 pre-adipocytes promotes adipogenesis through increased free cytosolic calcium (*Young et al., 2005*). In pre-neural stem cells, *Nnat* binds sarco/endoplasmic reticulum $Ca^{2+}$-ATPase (SERCA) to block $Ca^{2+}$ uptake into the ER thereby increasing cytosolic $Ca^{2+}$ levels (*Lin et al., 2010*). In addition to *Nnat*, *Hexb* regulates the uptake and accumulation of $Ca^{2+}$ in the ER via SERCA (*Pelled et al., 2003*). Upon the arrival of a signal, *F2r* (coagulation factor II receptor) and *Plcd1* (1-phosphatidylinositol 4,5-bisphosphate phosphodiesterase delta-1) in the transduction group initiate the adipogenesis cellular program. *F2r* is a G-protein-bound receptor that promotes phosphoinositide hydrolysis (*Soh et al., 2010*). Variation in the human F2R gene is associated with obesity (*Kichaev et al., 2019*). G-protein-coupled receptors transmit external signals into the cell where they are then propagated by secondary messenger systems, one of which is mediated by *Plcd1* (*Nakamura et al., 2005*; *McDonald and Mamrack, 1995*). The downstream effect of *Plcd1*-mediated signaling is the efflux of calcium into the cytosol from the ER, thereby increasing cytosolic $Ca^{2+}$ levels (*Thatcher, 2010*; *Berridge, 2016*). Increased cytosolic $Ca^{2+}$ in pre-adipocytes promotes phosphorylation of cAMP-response element-binding protein (CREB), which promotes activity of CCAAT/enhancer-binding protein (C/EBP) transcription factors, activating adipogenesis, altering the expression of *Cdkn1c* (cyclin dependent kinase inhibitor

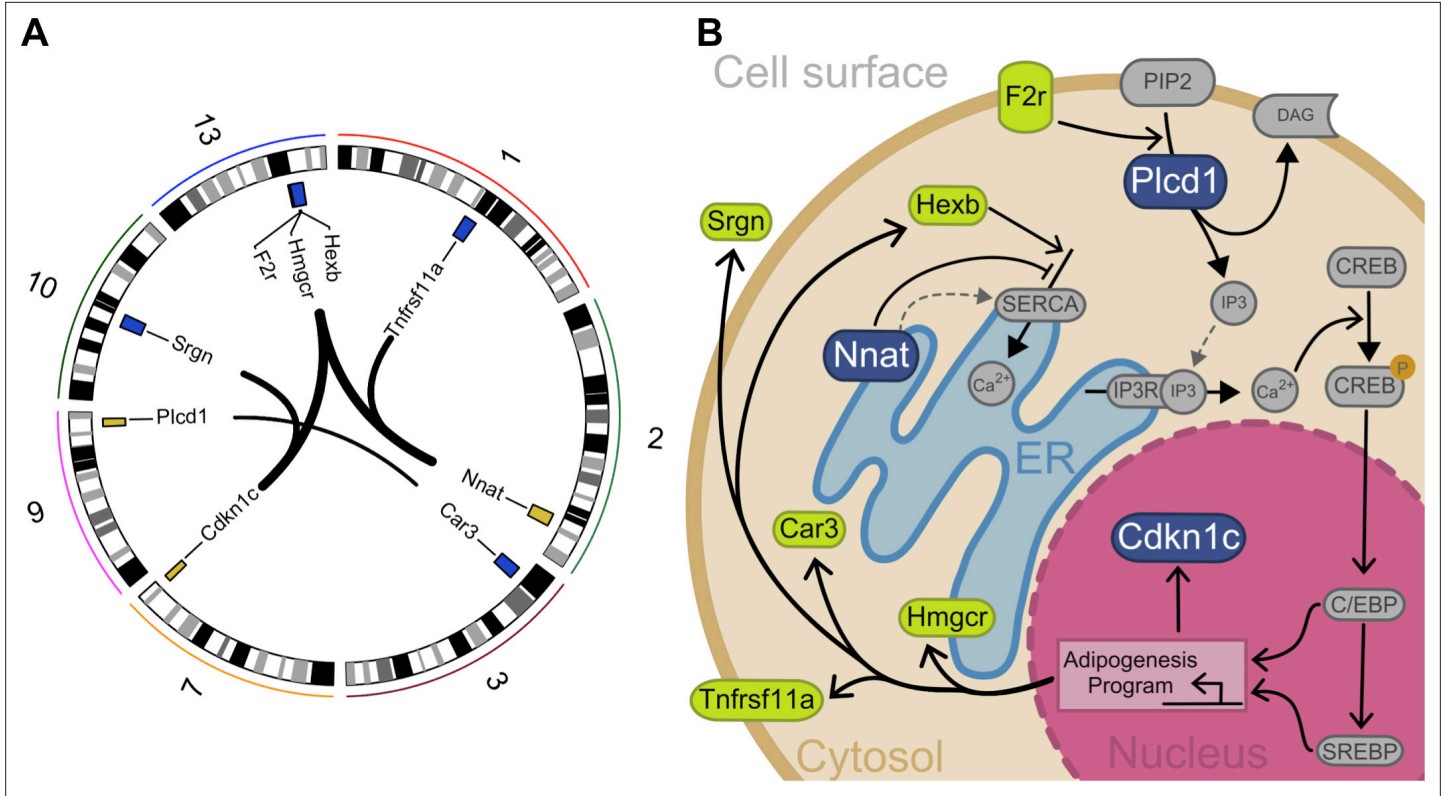

**Figure 3.** Interacting genes form a diet-responsive network affecting adipogenesis.
(**A**) There are nine significant imprinting-by-imprinting epistatic ASE/DE/phenotype sets in the $F_{16}$ advanced intercross population (n = 1002). Interactions are shown as lines connecting ASE (yellow) and DE genes (purple). Chromosome number is shown around the plot. (**B**) The epistatic parent-of-origin effect network is comprised of key steps in a putative pathway regulating differentiation and survival of adipocytes. This pathway was constructed by incorporating previously published cellular functions. The pathway members are color coded in blue for ASE genes (*Plcd1*, *Nnat*, and *Cdkn1c*) and green for DE genes (*F2r*, *Hexb*, *Hmgcr*, *Car3*, *Tnfrsf11a*, and *Srgn*). The network breaks down into potentiation, transduction, and response. *Nnat* and *Hexb* potentiate signaling by managing availability and accumulation of calcium necessary for signal transduction. Once a signal is received, *F2r* and *Plcd1* transduce it by activating second messengers to initiate a response. This response initiates an adipogenesis cellular program that affects expression of *Cdkn1c*, *Hmgcr*, *Car3*, *Tnfrsf11a*, and *Srgn*.

The online version of this article includes the following figure supplement(s) for figure 3:

**Figure supplement 1.** Example transformation of $F_{16}$ phenotypes.

**Figure supplement 2.** Stable null permutations plot for epistasis.

**Figure supplement 3.** Representative multiple tests correction of imprinting:imprinting epistasis.

1 C), *Hmgcr* (3-hydroxy-3-methylglutaryl-CoA reductase), *Car3* (carbonic anhydrase 3), *Tnfrsf11a* (TNF receptor superfamily member 11 a), and *Srgn* (serglycin).

 *Cdkn1c* is a canonically imprinted maternally expressed gene that inhibits cell proliferation (*Kang et al., 2008*). Increased expression of *Cdkn1c* is protective against diet-induced obesity in mice (*Van de Pette et al., 2018*), and in humans increased caloric intake results in decreased CDKN1C expression (*Franck et al., 2011*). *Hmgcr* is the rate-limiting enzyme in cholesterol biosynthesis (*Burg and Espenshade, 2011*; *Jo and Debose-Boyd, 2010*) and converts HMG-CoA into mevalonate, which is essential for adipocyte survival (*Yeh et al., 2018*). *Srgn* is an adipocytokine thought to be part of a feedback loop with *Tnfα* (tumor necrosis factor alpha), mediating paracrine cross-talk between macrophages and adipocytes (*Lemire et al., 2007*; *Imoto-Tsubakimoto et al., 2013*; *Schick et al., 2001*; *Zernichow et al., 2006*). *Srgn* is known to play a role in osteoblast-mediated bone mineralization (*Bigdeli et al., 2010*), which along with osteoclast-driven bone deconstruction drives bone remodeling (*Aubin, 1992*). Osteoblasts share a lineage with adipocytes, and the quantity of osteoblasts is inversely proportional to that of marrow adipose tissue (*Rodríguez et al., 2008*; *Prockop, 1997*; *Ali et al., 2005*; *Akune et al., 2004*; *Cho et al., 2011*; *Rosen and Bouxsein, 2006*; *Turner et al., 2018*).

*Tnfrsf11a* is a cell surface protein that regulates differentiation of osteoclasts (*Nakagawa et al., 1998*). Osteoprotegerin (OPG) is a decoy receptor for TNFRSF11A thereby inhibiting osteoclastogenesis and bone resorption (*Matsuo et al., 2020*). OPG is expressed during differentiation of 3T3L1 adipocytes (*An et al., 2007*). Expression of OPG is induced by *Tnfα* in 3T3L1 adipocytes and is associated with obesity in humans (*Holecki et al., 2007*; *Erol et al., 2016*; *Zaky et al., 2022*).

The exact function of OPG/*Tnfrsf11a* outside of osteoclastogenesis is unknown, but the function of osteoclasts is to break down bone tissue during bone resorption. Bone resorption regulates the level of blood calcium. The bioavailability of calcium in the blood potentially alters ER calcium stores, creating cross-talk between bone cells and white adipose tissue calcium signaling. Osteoclasts break down bone by acidifying mineralized bone, orchestrated by osteoblasts that have become embedded in the matrix they produce (osteocytes). Oxidative stress on osteocytes from the bone acidification process is prevented by *Car3*. *Car3* is an enzyme that catalyzes the conversion of carbonic acid to $CO_2$ and water. Its expression in white adipose is negatively correlated with, and responsive to, long-term obesity in mice and humans (*Stanton et al., 1991*; *Font-Clos et al., 2017*). *Car3* does not protect against diet-induced obesity and is not necessary for fatty acid synthesis (*Renner et al., 2017*). As such its exact function in adipocytes is unknown.

## Nnat and F2r covary in white adipose tissue and their interaction associates with variation in basal glucose levels across generations

To better understand how these interactions affect phenotype, we focused on the negative correlation of the imprinted gene, *Nnat,* and the biallelic gene, *F2r,* in the above network in high fat-fed females, the cohort with the most significant covariation in the $F_1$ animals (FDR = 0.05). *Nnat* and *F2r* show significant imprinting-by-imprinting epistasis for basal glucose levels in the $F_{16}$ population (FDR = $6.00e^{-16}$; *Figure 4A and B*). To validate gene expression patterns, we combined $F_1$ biological replicates and $F_0$ high fat-fed female animals ($F_1$ n = 13 and $F_0$ n = 12) and again observe that *F2r* and *Nnat* are each significantly differentially expressed between reciprocal heterozygotes, that is by cross (*Figure 4C and D*). Further, the co-expression of *Nnat* and *F2r* also persists in the $F_0/F_1$ population (*Figure 4E*).

A limitation of identifying covariation patterns in $F_1$ and $F_0$ populations is that all loci are linked. This makes it difficult to determine which ASE genes truly co-express with DE genes. While incorporation of orthogonal $F_{16}$ genotypes and phenotypes helps reduce false discoveries, a population with randomized genetic background for which we have expression data is needed to replicate these results. To that end, $F_2$ animals were generated and *Nnat* and *F2r* gene expression levels were measured via qPCR (n = 14). We found that *F2r* and *Nnat* are significantly co-expressed in high-fat-fed female $F_2$ animals (*Figure 4H*).

*F2r* expression significantly positively correlates with basal glucose levels in the RNA-sequenced high-fat-fed female $F_1$ animals (r = 0.514, FDR = 0.01; *Supplementary file 5*). *F2r* expression is also significantly positively correlated with basal glucose in the combined $F_0/F_1$ population (*Figure 4G*). A negative trend between *Nnat* expression and basal glucose level is observed but not statistically significant in the combined $F_0/F_1$ animals (*Figure 4F*). Correlation of *F2r*'s and *Nnat*'s individual expression with basal glucose in $F_2$ mice follows the same pattern as in the $F_0/F_1$'s. Bootstrapping to calculate confidence intervals shows that the correlation differences between $F_0/F_1$ and $F_2$ are not significant (*Figure 4I and J*; *Figure 4—figure supplement 1*). However, the product of *Nnat* and *F2r* expression (*Nnat* x *F2r*) is significantly predictive of basal glucose (p = 0.045, $R^2$ = 0.29). This indicates that expression of *Nnat* and *F2r*, as a function of their genotypes and allelic parent-of-origin, are not individually sufficient to explain variation in basal glucose levels. But together they are able to explain a significant amount of phenotypic variation. This is precisely what our epistatic model would predict.

Finally, studying the $F_2$ animals allows us to determine if maternal mitochondrial ancestry contributes significantly to *Nnat* or *F2r* expression or to variation in basal glucose. We find mitochondrial genome identity does not significantly covary with *F2r* expression (p = 0.198), *Nnat* expression (p = 0.365), or basal glucose (p = 0.388).

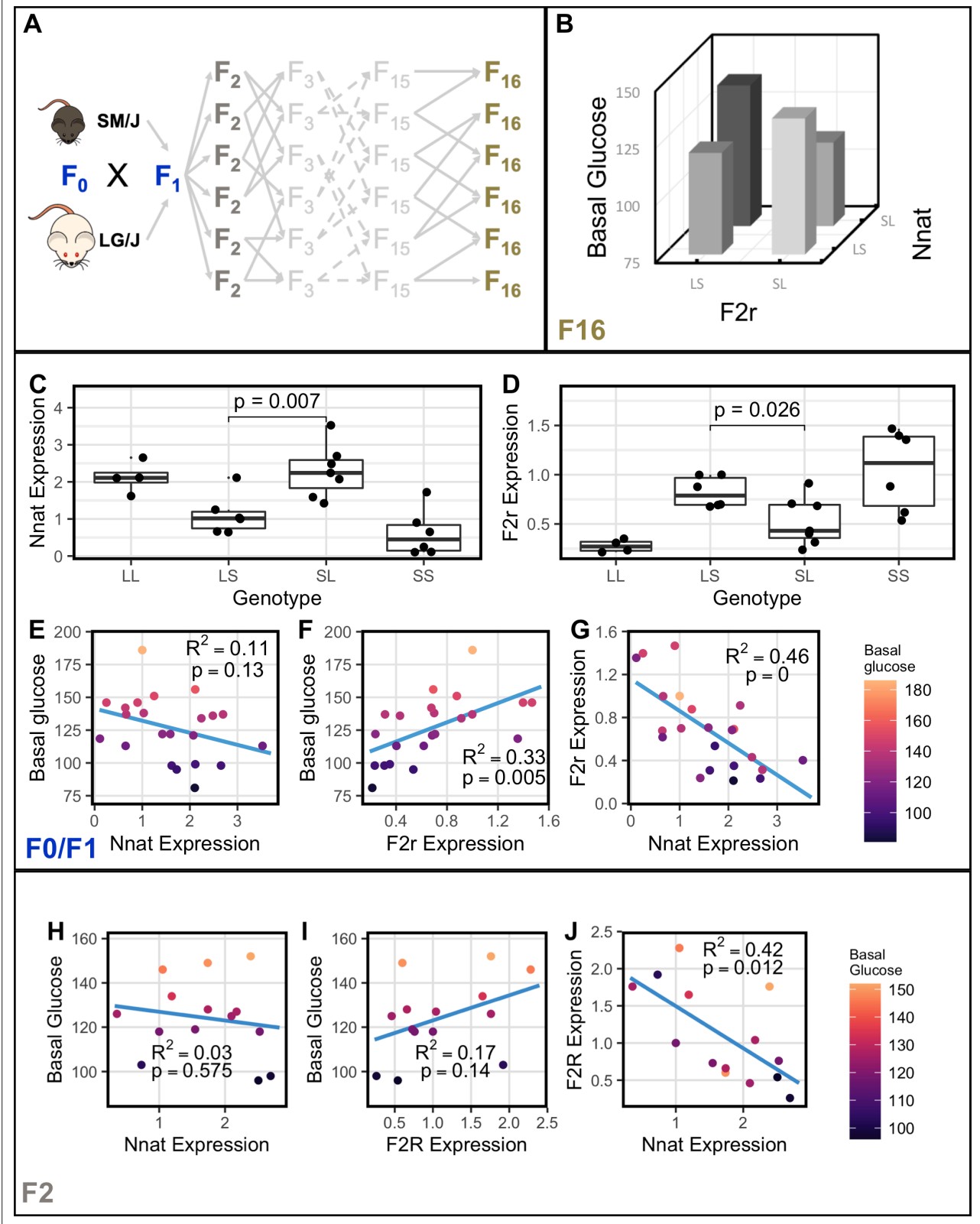

**Figure 4.** *Nnat* and *F2r* covary across generations.

(**A**) Breeding scheme for the $F_{16}$ Advanced Intercross between the LG/J and SM/J inbred strains. (**B**) Significant imprinting-by-imprinting epistasis associated with variation in basal glucose (n = 1002). The parent-of-origin effects of *F2r* on basal glucose depend on the parent-of-origin effects at *Nnat*. (**C**) Expression of *Nnat* across genotypes in a combined $F_0/F_1$ population of high fat-fed females (n = 25). (**D**) Expression of *F2r* across genotypes

*Figure 4 continued on next page*

*Figure 4 continued*

in a combined F$_0$/F$_1$ population of high-fat-fed females (n = 25). (**E**) Significant correlation between *Nnat* and *F2r* expression in the F$_0$/F$_1$ mice (F$_1$ n = 13; F$_0$ n = 12). (**F**) and (**G**) Correlations between basal glucose and *Nnat* and *F2r* in the F$_0$/F$_1$ mice (F$_1$ n = 13; F$_0$ n = 12). (**H**) Significant correlation between *Nnat* and *F2r* expression in the high fat-fed female F$_2$ mice (n = 14). (**I**) and (**J**) Correlations between basal glucose and *Nnat* and *F2r* are not individually significant in the F$_2$ mice. Alleles are ordered maternal | paternal within the genotype classes.

The online version of this article includes the following figure supplement(s) for figure 4:

**Figure supplement 1.** Pearson's correlation coefficient confidence intervals.

## Single-cell RNAseq reveals that Nnat expression increases and F2r expression decreases in pre-adipocytes along an adipogenic trajectory

To determine what cell types express *Nnat* and *F2r* and whether the directionality of the *Nnat* imprinted → *F2r* target correlation persists along the adipogenic trajectory, we turned to single-cell RNAseq. We used publicly available scRNAseq data collected from stromal vascular cells isolated from C57BL/6 J epididymal adipose tissue (*Burl et al., 2018*). Cell type identity was assigned using previously reported markers for this data set (*Adipoq* = differentiating mesenchymal stem cells; *Pdgfra* = mesenchymal stem cells; *Csf1r* = macrophage; *Cdh5* = vascular endothelial cells; *Acta2* = vascular smooth muscle cells; *Cd2* = B cells) (*Supplementary file 7*; *Figure 5—figure supplement 1*). The adipogenic trajectory refers to cells transitioning from pre-adipocytes (mesenchymal stem cells) to cells differentiating into adipocytes. Clusters along this trajectory were identified by the opposing expression patterns of *Pdgfra* and *Adipoq* (*Figure 5A-D and I*). We found that *Nnat* expression increases along the trajectory while F2r expression decreases (*Figure 5E–F and H*). Further there

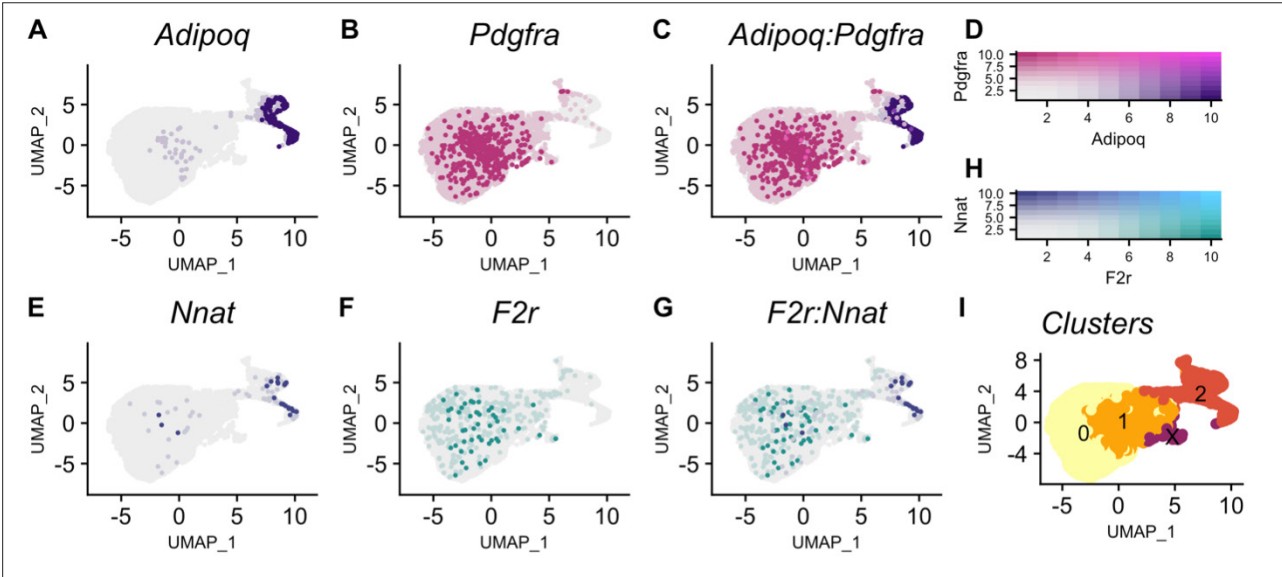

**Figure 5.** *Nnat* expression increases and *F2r expression decreases* in pre-adipocytes along an adipogenic trajectory.
(**A**) *Adipoq* is a marker of adipocytes whose expression (purple) increases along the trajectory. (**B**) *Pdgfra* is a marker of mesenchymal stem cells whose expression (pink) decreases along the trajectory. (**C**) Cells in clusters expressing one or both *Adipoq* and *Pdgfra* fall along an adipogenic trajectory. (**D**) Intensity of expression of *Adipoq* and *Pdgfra* indicated by coloration. (**E**) *Nnat* expression (blue) increases along the trajectory. (**F**) *F2r* expression (teal) decreases along the trajectory. (**G**) Negative association between *Nnat* and *F2r* expression within adipocytes along the trajectory. (**H**) Intensity of expression of *Nnat* and *F2r* indicated by coloration. (**I**) The adipogenic trajectory is broken into subclusters of cells with no *Adipoq* expression (cluster 0) to high *Adipoq* expression (cluster 2).

The online version of this article includes the following figure supplement(s) for figure 5:

**Figure supplement 1.** Cell types were defined using canonical markers.

**Figure supplement 2.** Candidate genes are differentially expressed across adipogenic trajectory.

**Figure supplement 3.** Single cell quality was controlled.

**Figure supplement 4.** Determining the resolution for clustering.

**Figure supplement 5.** Adipocyte clustering resolution was selected to minimize Adipoq variation.

is a negative association between Nnat and F2r expression within adipocytes along the trajectory (*Figure 5G*). This pattern is consistent with the negative correlation we observe between Nnat and F2r in the bulk white adipose tissue. Because available scRNAseq data do not match the exact sex/ diet/ genetic background contexts of the LGxSM mice, there will be unaccounted for differences between the data sets. However, the observed consistent pattern indicates that the pathway structure persists across sex/ diet/ genetic backgrounds.

In addition to interrogating *Nnat* and *F2r* in single cells along an adipogenic trajectory, we found that eight of the nine genes comprising the epistatic parent-of-origin effect network described above are differentially expressed along the trajectory, and they associate with cell types that are consistent with their respective roles in adipose tissue (*Figure 5—figure supplement 2*).

## Discussion

Epistatic interactions between imprinted and non-imprinted genes can influence complex traits when the genotypic effects of one gene depends on the parent-of-origin of alleles at another (*Lawson et al., 2013*; *Wolf and Cheverud, 2009*). Here, we examined epistatic interactions associated with parent-of-origin effects on dietary-obesity traits in white adipose using a simple yet powerful $F_1$ reciprocal cross mouse model. Although these parent-of-origin dependent allele-specific expression biases are consistent with imprinting mechanisms, we cannot rule out that maternal and/or paternal effects also contribute to the phenomena we observe (*Hager et al., 2008*).

Interactions between imprinted and non-imprinted genes have previously been shown to contribute to variation in metabolic phenotypes. For example, the maternally expressed transcription factor KLF14 (kruppel-like factor 14) regulates biallelic gene expression related to adiposity (*Parker-Katiraee et al., 2007*; *Small et al., 2011*). Mapping studies have identified two SNPs (rs4731702, rs972283) upstream of KLF14 associated with type II diabetes and cholesterol levels (*Voight et al., 2010*; *Teslovich et al., 2010*). Both variants have maternally restricted *cis*-regulatory associations with KLF14 expression in adipose tissue (*Kong et al., 2009*). eQTL analysis found that rs4731702 is also enriched for *trans*-associations with KLF family transcription factor binding sites in subcutaneous white adipose tissue, suggesting that KLF14 may be a master transcriptional regulator in adipose tissue (*Small et al., 2011*). Whether additional pairs of imprinted and biallelic genes are similarly co-expressed and associate with phenotypic variation remains an open question that has not been thoroughly investigated in large landmark functional genomics studies including ENCODE, GTEx, and GWAS, leaving a significant gap in our knowledge. Interactions between imprinted and biallelic genes could explain some of the observed parent-of-origin effect patterns associated with regions lacking obvious candidate genes, as described in a recent survey of 97 complex traits measured in outbred mice (*Mott et al., 2014*).

Our model asserts that parent-of-origin effects start at ASE genes and are transduced through DE genes onto phenotype. This is illustrated in the interaction between *Nnat* and *F2r*. If a *cis*-regulatory effect interacts with epigenetic modifications (i.e. imprinting) at *Nnat*, then *Nnat* expression of genotypic classes are affected by paternal allele identity (*Lawson et al., 2013*). Between the LG/J and SM/J alleles at *Nnat*, the LG/J allele is more highly expressed. If our model is correct, the downstream DE gene should show a corresponding pattern (*Figure 1B*). In the case of *Nnat* and *F2r*, which have strong negative correlated expression, when the LG/J allele is inherited paternally at *Nnat*, the higher expression of *Nnat* should correspond with lower expression of *F2r*. This is what we observe (*Figure 4*). If this relationship is true, we should see persistent co-expression of *Nnat* and *F2r* across genetic backgrounds ($F_0$, $F_1$, $F_2$), which we do (*Figure 4*). This supports a biologically meaningful relationship between *Nnat* and *F2r*. Our model further predicts that the DE genes should more closely affect phenotype (*Pierce et al., 2014*; *Shan et al., 2019*; *Lutz and Hokanson, 2015*). In the case of *Nnat* and *F2r*, we expect *F2r* to more strongly associate with basal glucose levels than *Nnat*, which we observe (*Figure 4*).

There is a clear relationship between *Nnat* and *F2r* in adipogenesis, but the specifics of how this relationship extends to glucose homeostasis are unclear. One possibility is that by altering SERCA function, *Nnat* affects not only the formation of new adipocytes, but also the beiging of adipocytes. The SERCA channel is uncoupled in beige adipocytes as part of a UCP1-independent form of non-shivering thermogenesis (*Ikeda and Yamada, 2020*). Non-shivering thermogenesis consumes a significant amount of energy, thereby altering glucose homeostasis (*Carson et al., 2020*). This hypothesis

links these genes to physiological processes that are consistent with evolutionary hypotheses about the prevalence of parent-of-origin effects. *Nnat* and *F2r* are members of a putative network we identified that is coordinated by interactions between ASE and DE genes. From the literature, we found that the genes in this epistatic network function in key steps in a pathway regulating differentiation and survival of adipocytes in response to nutritional environment (*Figure 3B*). Specifically, there is evidence that it plays a critical role in the induction of adipogenesis. This alone demonstrates how parent-of-origin effects can move through networks along molecular pathways. Beyond proof-of-principle this network provides a clue to the puzzle of the prevalence of parent-of-origin effects.

The constituents of this single network appear to play vastly different physiological roles depending on the tissue. In white adipose the network appears to play some role in balancing proliferation, differentiation, and apoptosis as we describe above. In pancreatic ß-cells, members of this network affect insulin secretion (*Millership et al., 2018*). In bone, members of this network affect the balance of cartilage/bone growth and reabsorption. These three physiological processes may at first seem unrelated, but they share one key commonality – they are jointly critical to growth. This is consistent with the sexual conflict hypothesis attributed to parent-of-origin effects (*Patten et al., 2014*; *Babak et al., 2015*). The of size of progeny in placental mammals can have opposing fitness consequences for mothers/ maternal relatives and fathers/ paternal relatives. The fitness of fathers and paternal relatives, particularly in the case of multi-paternity litters, is improved with larger progeny (*Mochizuki et al., 1996*; *Babak et al., 2015*; *Fowden and Moore, 2012*; *Patten et al., 2014*; *Wilkins and Haig, 2003*; *Haig, 1997*). This comes at a fitness disadvantage to the mother. The fitness of mothers is improved by progeny of a manageable size, allowing her to produce multiple litters.

According to this model, imprinting evolved in part to allow one parental lineage to hijack parts of a nutritional environment response pathway driving growth in a direction favorable to maximize the fitness of that lineage. Key processes in such a pathway driving growth would include the secretion of growth factors, construction of cartilage and bone, and the accumulation of energy stores. We present a network that appears to play a role in all three processes. If the sexual conflict hypothesis is true, then the most parsimonious place for imprinting to evolve would be in key regulatory points that affect as many aspects of growth as possible. This is consistent with the network we identified, a single pathway affecting many aspects of growth. This hints at the possibility that parent-of-origin effects are common because of the multi-purpose nature of the pathways in which genomic imprinting manifests and parent-of-origin effects propagate.

By leveraging the reciprocal $F_1$ hybrids, we are able to integrate parent-of-origin-dependent allele specific expression and parent-of-origin-dependent differential expression with $F_{16}$ phenotypes. By doing so, we identify plausible candidates for functional validation and describe discrete molecular networks that may contribute to the observed parent-of-origin effects on metabolic phenotypes. The genes and interactions we present here represent a set of actionable interacting candidates that can be probed to further identify the machinery driving these phenomena and make predictions informed by genomic sequence. The frameworks we have developed account for the genetic, epigenetic, and environmental components underlying these parent-of-origin effects, thereby improving our ability to predict complex phenotypes from genomic sequence. We focused on metabolic phenotypes in this study, but the patterns we identified may translate to other complex traits where parent-of-origin effects have been implicated.

# Materials and methods

**Key resources table**

| Reagent type (species) or resource | Designation | Source or reference | Identifiers | Additional information |
|---|---|---|---|---|
| Other | High fat diet | Teklad | TD88137 | 42% kcal from fat |
| Other | Low-fat diet | Research Diets | D12284 | 15% kcal from fat |
| Commercial assay, kit | RNeasy Lipid Tissue Kit | QIAgen | 74,804 | |
| Commercial assay, kit | RiboZero kit | Illumina | 20040529 | |
| Commercial assay, kit | DNA 1000LabChip | Agilent | 5067–1504 | |

*Continued on next page*

*Continued*

| Reagent type (species) or resource | Designation | Source or reference | Identifiers | Additional information |
|---|---|---|---|---|
| Commercial assay, kit | High-Capacity cDNA Reverse Transcription Kit | Thermo Fisher | 4368814 | |
| Sequence-based reagent | *Nnat* forward primer | This paper | | Detailed information is found in the methods section |
| Sequence-based reagent | *Nnat* reverse primer | This paper | | Detailed information is found in the methods section |
| Sequence-based reagent | *F2r* forward primer | This paper | | Detailed information is found in the methods section |
| Sequence-based reagent | *F2r* reverse primer | This paper | | Detailed information is found in the methods section |
| Sequence-based reagent | *L32* forward primer | This paper | | Detailed information is found in the methods section |
| Sequence-based reagent | *L32* reverse primer | This paper | | Detailed information is found in the methods section |
| Software, algorithm | R | R | 3.6.1 | |
| Software, algorithm | STAR | STAR | DOI: 10.1093/bioinformatics/bts635 | |
| Software, algorithm | FASTQC | FASTQC | other | https://www.bioinformatics.babraham.ac.uk/projects/fastqc/ |
| Software, algorithm | EdgeR | CRAN | DOI: 10.1093/bioinformatics/btp616 | |
| Software, algorithm | WEB-based Gene Set Analysis Toolkit | WEB-based Gene Set Analysis Toolkit | DOI: 10.1093/nar/gkz401 | |
| Software, algorithm | Seurat | Seurat | DOI: 10.1038/nbt.3192 | |
| Strain, strain background (Mus musculus) | SM/J | The Jackson Laboratory | 000687 | |
| Strain, strain background (*Mus musculus*) | LG/J | The Jackson Laboratory | 000675 | |

## Mouse husbandry and phenotyping

LG/J and SM/J founders ($F_0$) were obtained from The Jackson Laboratory (Bar Harbor, ME). $F_1$ reciprocal cross animals were generated by mating LG/J mothers with SM/J fathers (LxS) and the inverse (SxL). $F_2$ intercrossed animals were generated by mating LxS mothers with SxL fathers and the inverse. After weaning at 21 days, animals were separated into sex-specific cages of 3–5 animals per cage and randomly placed on high-fat (42% kcal from fat; Teklad TD88137) or low-fat (15% kcal from fat; Research Diets D12284) isocaloric diets. Feeding was ad libitum. There were 96 experimental $F_1$ animals in total, with 48 animals for each cross (LxS and SxL). Within each cross, there were 24 high-fat-fed animals (12 males; 12 females) and 24 low-fat-fed animals (12 males; 12 females). The $F_2$ animals were generated for a different study, following the same weaning protocol and diets, and we used data from the high fat-fed females (n = 14) for validation in the the current study (*Carson et al., 2020*). Additionally, we used data generated from founder $F_0$ (LG/J (n = 6) and SM/J (6)) high fat-fed female animals, also generated for a different study and subjected to the same weaning protocol and diets (*Carson et al., 2020*). The barrier animal facilities at WUSM follow a 12/12 hr light/dark schedule, all water is autoclaved and changed weekly, and all cages are changed weekly.

All animals were weighed weekly from three weeks of age until sacrifice. At 19 weeks of age, body composition was determined by MRI and a glucose tolerance test was performed after a 4 hr fast. At 20 weeks of age, animals were given an overdose of sodium pentobarbital after a 4 hr fast and blood was collected via cardiac puncture. Euthanasia was achieved by cardiac perfusion with phosphate-buffered saline. After cardiac perfusion, the reproductive fat pad was harvested, flash frozen in liquid nitrogen, and stored at –80 °C.

## Study design

The weaning, phenotyping protocols, and diets were chosen to reproduce the protocols and diets used in studies of the F16 Advanced Intercross of the LG/J x SM/J inbred mouse lines that were used in previously published mapping studies (*Cheverud et al., 2011*; *Lawson et al., 2011a*; *Lawson et al., 2011b*, *Lawson et al., 2010*). The experimental unit for the current study is the individual mouse. For the RNA sequencing, a single animal was randomly chosen from each cage using a random number generator in R. All other animals served as biological replicates. Mice from multiple cages representing different crosses, generations, diets, and sexes, were necropsied at the same time to avoid batch effects. Library prep and RNA sequencing was performed blinded by the WUSM Genome Technology and Access Center.

## Genomes and annotations

LG/J and SM/J indels and SNVs were leveraged to construct strain-specific genomes using the GRC38.72-mm10 reference as a template (*Nikolskiy et al., 2015*). This was done by replacing reference bases with alternative (LG/J | SM/J) bases using custom python scripts. Ensembl R72 annotations were adjusted for indel-induced indexing differences for both genomes.

## RNA sequencing

Total RNA was isolated from adipose tissue using the RNeasy Lipid Tissue Kit (QIAgen) (n = 32, 4 animals per sex/diet/cross cohort). RNA concentration was measured via NanoDrop and RNA quality/integrity was assessed with a BioAnalyzer (Agilent). RNA-Seq libraries were constructed using the RiboZero kit (Illumina) from total RNA samples with RIN scores > 8.0. Libraries were checked for quality and concentration using the DNA 1000LabChip assay (Agilent) and quantitative PCR, according to manufacturer's protocol. Libraries were sequenced at 2 × 100 paired end reads on an Illumina HiSeq 4,000. After sequencing, reads were de-multiplexed and assigned to individual samples. RNAseq data are available through the NCBI-SRA, accession: PRJNA753198.

## Library complexity

Complexity was measured by fitting a beta-binomial distribution to the distribution of $L_{bias}$ values using the VGAM package (*Yee, 2010*). The shape parameters (α, β) of beta-binomial distributions were estimated and used to calculate dispersion ( $\rho$ ). Dispersion values less than 0.05 indicate our libraries are sufficiently complex (*Figure 2—figure supplement 1*).

$$\rho_s = \frac{1}{1+\alpha_s+\beta_s}$$

One library was found to have insufficient complexity and was removed from the analyses.

## Allele-specific expression

FASTQ files were filtered to remove low quality reads and aligned against both LG/J and SM/J pseudo-genomes simultaneously using STAR with multimapping disallowed (*Dobin et al., 2013*). Read counts were normalized via upper quartile normalization and a minimum normalized read depth of 20 was required. Alignment summaries are provided in *Supplementary file 8* and *Figure 2—figure supplement 2*.

For each gene in each individual, allelic bias ($L_{bias}$) was calculated as the proportion of total reads for a given gene with the LG/J haplotype. Parent-of-origin-dependent allele-specific expression was detected by ANOVA using one of a number of models in which $L_{bias}$ is responsive to cross and the interaction of cross with some combination of sex and diet:

$$\text{Model} \begin{cases} \textit{if each Cross context has} \geq 2 \textit{ samples, } Lbias \sim Cross \\[6pt] \textit{if each Cross}: \textit{Sex context has} \geq 2 \textit{ samples, } Lbias \sim Cross + Cross: Sex \\[6pt] \textit{if each Cross}: \textit{Diet context has} \geq 2 \textit{ samples, } Lbias \sim Cross + Cross: Diet \\[6pt] \textit{if each context has} \geq 2 \textit{ samples, } Lbias \sim Cross + Cross: Sex + Cross: Diet + Cross: Sex: Diet \end{cases}$$

Accurately estimating the significance of these effects and correcting for multiple tests is challenging for two reasons: (1) the complexity of the many environmental contexts and (2) the correlation of allelic bias within and between imprinted domains breaks assumptions of independence. A permutation approach is an effective way to overcome these challenges. The context data was randomly shuffled for each gene independently and analyses were rerun to generate a stable null distribution of p-values (*Figure 2—figure supplement 3*). False discovery rates were estimated for a given significance threshold as the proportion of significant tests under the permutated null model relative to significant tests under the real data model. A value of 1 meaning that 100% of tests at a given significance threshold are likely false positives. An FDR ≤ 0.1 was considered significant (*Supplementary file 1*, *Figure 2—figure supplement 4*).

To determine the parental direction and size of expression biases, a parent-of-origin effect score was calculated as the difference in mean $L_{bias}$ between reciprocal crosses (LxS or SxL). Parent-of-origin effect scores range from completely maternally expressed (–1), to biallelic (0), to completely paternally expressed ( + 1). Parent-of-origin effect score thresholds were calculated from a critical value of $\alpha =$ 0.01, determined from a null distribution created by permutation Genes with significant allele-specific expression and parent-of-origin scores beyond the critical value were considered to have significant parent-of-origin-dependent allele-specific expression (*Figure 2—figure supplement 5*).

## Differential expression

Differential expression by reciprocal cross was determined by first aligning reads against the LG/J and SM/J genomes simultaneously with multimapping permitted. Reads were normalized by Trimmed mean of M-values (TMM) normalization, which estimates scale factors among samples to allow for differences in RNA composition (*Robinson and Oshlack, 2010*). A minimum normalized read count of 10 was required. Generalized linear models accounting for diet, sex, and diet-by-sex were fit in EdgeR (*Robinson et al., 2010*). Differential expression was detected by a likelihood ratio test. Significance was determined for five models for each gene:

$$1.\, Expression \,\sim\, Cross$$

$$2.\;\; Expression \,\sim\, Cross:Sex$$

$$3.Expression \,\sim\, Cross:Diet$$

$$4.\; Expression \,\sim\, Cross:Sex:Diet$$

$$5.\; Expression \,\sim\, Cross + Cross:Sex + Cross:Diet + Cross:Sex:Diet$$

Multiple test corrections were performed by implementing the 'qvalue' R package to estimate false discovery rates (*Figure 2—figure supplement 6*). Genes with a FDR of ≤0.1 and a $|fold\,change| \geq 1.5$ were considered significantly differentially expressed by reciprocal cross (*Figure 2—figure supplement 7* and *Supplementary file 2*).

## Gene-gene interactions

Networks were constructed in each tissue by pairing genes showing parent-of-origin-dependent allele-specific expression with biallelic genes that are differentially expressed by cross. Pairs were predicted by modeling the expression of biallelic genes as a function of parent-of-origin-dependent allele-specific expression, $L_{bias}$, sex, diet, and sex-by-diet. The strength of a prediction was measured through model fit, which was estimated as a mean test error with 10-fold cross-validation employed to prevent overfitting. Significance was estimated by likelihood ratio test using a chi-square distribution. Given the complexity of contexts, false discovery rates were determined by permuting the context and expression data to generate a stable null-distribution of p-value (*Figure 2—figure supplement 8*) Null distribution stability was evaluated by calculating the critical value for alpha = 0.05 at each genome wide iteration. The standard deviation of critical values was calculated after each iteration for the last 5 iterations. Genome-wide shuffling was done 500 times, with 759 independent randomized tests per iteration, meaning the stable null model is composed of 379,500 randomized observations. Using the null model, the 'qvalue' package estimated a $\hat{\pi}_0$ . This estimate was then used to estimate false discovery rates in the real data. MTE score thresholds were calculated from a critical value of $\alpha =$ 0.01, determined from a null distribution created by permutation (*Figure 2—figure supplement 9*).

Connections with an FDR ≤ 0.1 (*Supplementary file 9*) and MTE below the critical value were considered significant (*Figure 2—figure supplement 10*).

## Functional enrichment analysis

Functional enrichment of interacting genes showing parent-of-origin-dependent allele-specific expression with biallelic genes that are differentially expressed by cross was tested by over-representation analysis in the WEB-based Gene Set Analysis Toolkit v2019 (*Zhang et al., 2005*). We performed analyses of gene ontologies (biological process, cellular component, and molecular function), pathway (KEGG), and phenotype (Mammalian Phenotype Ontology). The list of all unique interacting genes was analyzed against the background of all unique genes expressed in white adipose. A Benjamini-Hochberg FDR-corrected p-value ≤ 0.01 was considered significant (*Supplementary file 4*).

## Phenotype correlation

In order to identify which phenotypes might be affected by genes in the parent-of-origin effects network, gene expression was correlated with metabolic phenotypes collected for $F_1$ animals with the contexts combined. Phenotypes were log transformed when necessary, as determined by Shapiro Wilkes test to approximate normality (*Figure 2—figure supplement 11*). Additionally, the effects of sex and diet were residualized out leaving only the effect of cross. This was done to mirror later residualizing of phenotypes in the F16 population when testing for epistasis. Multiple test corrections were performed by implementing the 'qvalue' R package to estimate false discovery rates (*Figure 2—figure supplement 12*). The minimum Pearson's correlation coefficient threshold was set to |0.5|. Connections with an FDR ≤ 0.05 (*Supplementary file 5*) and MTE below the critical value were considered significant (*Figure 2—figure supplement 13*).

## Epistasis testing

The $F_{16}$ LxS advanced intercross population, phenotypes, genotypes, genotypic scores, and QTL mapping methods are described elsewhere (*Cheverud et al., 2011*; *Lawson et al., 2011a*; *Lawson et al., 2011b*, *Lawson et al., 2010*). We tested for epistasis in interacting pairs between genes showing parent-of-origin-dependent allele-specific expression and biallelic genes that are differentially expressed by cross. We selected $F_{16}$ genotyped markers that fall within 1.5 Mb up- and downstream from the geometric center of each gene, defined as the genomic position halfway between the transcription start and stop position of that gene (*Supplementary file 10*). For every $F_{16}$ animal, an 'imprinting score' was assigned to each marker based on that animal's genotypic values (LL = 0, LS = 1, SL = –1, SS = 0; maternal allele is depicted first). Non-normally distributed phenotypes (as evaluated by a Shapiro-Wilk test) were $\log_{10}$-transformed to approximate normality (*Figure 3—figure supplement 1*). Because of the number of epistasis tests performed and the number of contexts represented in the data, we removed the effects of sex, diet and their interaction from each $F_{16}$ phenotype with a covariate screen. We tested for epistasis on the residualized data using the following generalized linear model:

$$R_{pheno} \sim \ BDE_{IMP} + ASE_{IMP} + BDE_{IMP} : ASE_{IMP}$$

where $R_{pheno}$ is the residual phenotype, $BDE_{IMP}$ is the imprinted genotypic score for the biallelic gene that is differentially expressed by cross, $ASE_{IMP}$ is the imprinted genotypic score for the gene showing parent-of-origin-dependent allele-specific expression bias, and $BDE_{IMP}:ASE_{IMP}$ is the interaction between the two genes' imprinted genotypic score. We employed a permutation approach to accurately estimate significance given the linkage of proximal markers. Imprinted genotypic values were randomly shuffled to generate a stable null model of p-values (*Figure 3—figure supplement 2*). False discovery rates were estimated for a given significance threshold as the proportion of significant tests under the permutated null model relative to significant tests under the real data model (*Figure 3—figure supplement 3*). An FDR ≤ 0.1 was considered significant. Epistasis was considered significant if the $BDE_{IMP}: ASE_{IMP}$ interaction term met the significance threshold (*Supplementary file 6*).

## Validation of Nnat and F2r expression patterns

Expression patterns of *Nnat* and *F2r* in white adipose were validated by qRT-PCR in high-fat-fed female LG/J and SM/J mice and in biological replicates of high-fat-fed female $F_1$ reciprocal cross

animals (n = 6 LG/J homozygotes, n = 10 LxS and 10 SxL reciprocal heterozygotes, n = 6 SM/J homozygotes). Total RNA was extracted from adipose samples using the Qiagen RNeasy Lipid Kit. High-Capacity cDNA Reverse Transcription Kit (Thermo Fisher) was used for reverse transcription. Quantitative RT-PCR was performed with an Applied Biosystems (USA) QuantStudio 6 Flex instrument using SYBR Green reagent. Results were normalized to *L32* expression using the ΔΔCt method. *Nnat* forward primer – CTACCCCAAGAGCTCCCTTT and reverse primer – CAGCTTCTGCAGGGAGTACC . *F2r* forward primer – TGAACCCCCGCTCATTCTTTC and reverse primer – CCAGCAGGACGCTTTC ATTTTT. *L32* forward primer – TCCACAATGTCAAGGAGCTG and reverse primer – GGGATTGGTGAC TCTGATGG. Data points were considered outliers if they led to violation of normality assumptions or were considered outliers by box and whisker plots. ANOVA was used to estimate significance of differential expression by cross (1), paternal allele identity (2), mitochondrial ancestry (3).

$$1.\ Expression \sim\ Cross\ \in\ \begin{cases} LL,\ 0 \\ LS,\ -1 \\ SL,\ 1 \\ SS,\ 0 \end{cases}$$

$$2.\ Expression \sim\ Paternal\ Allele\ \in\ \begin{cases} LL,\ 0 \\ LS,\ 1 \\ SL,\ 0 \\ SS,\ 1 \end{cases}$$

$$3.\ Expression \sim\ Mitochondrial\ ancestry\ \in\ \begin{cases} LxS\ x\ SxL,\ 0 \\ SxL\ x\ LxS,\ 1 \end{cases}$$

Expression patterns were also validated by qRT-PCR in high fat-fed female $F_2$ animals (n = 14). Co-expression was determined by fitting a general linear model and estimating significance using the Wald test approximation of the LR test. Correlation with basal glucose was determined by fitting a general linear model and estimating significance using the Wald test approximation of the LR test. Pearson's correlation coefficients were calculated for each gene with basal glucose. To test whether patterns in these correlations was significantly different between $F_0/F_1$ and $F_2$ populations, boot-strapping was used to calculate 90% confidence intervals for the Pearson's correlation coefficients. 5,000 iterations were run with 10 individuals randomly selected with replacement. scRNA analysis of Nnat and F2r scRNAseq data was downloaded from SRA: SRP145475 (*Burl et al., 2018*). Data were processed and aligned to the C57BL/6 J reference (mm10) using Cell Ranger (*Zheng et al., 2017*). Analysis and cell quality control was performed using the Seurat (3.2.2)(*Stuart et al., 2019*) package in R (3.6.1)(*R Development Core Team, 2013*). Cell quality was controlled using three metrics (*Luecken and Theis, 2019*): (1) number of features, (2) number of counts, (3) covariation of features and counts. High quality cells were required to have between 500 and 3000 features and read counts between 1000 and 30,000. As sequencing is a process of random sampling, the number of features and the number of counts should covary. This relationship was fit to a generalized additive model. Deviation from this relationship (residuals) were computed for each cell. High-quality cells were required to have a residual within 3 standard deviations of the mean residual of all cells (*Figure 5—figure supplement 3*).

Seurat normalization with a scale factor of 10,000 was performed. Dimensionality reduction (UMAP) was performed (dims = 1:10, resolution = 0.15). Resolution was chosen using the clustree (0.4.3) package (*Zappia and Oshlack, 2018*). A range of resolutions from 0.06 to 0.18 were tested, and the highest resolution with stable clustering was chosen (*Figure 5—figure supplement 4*). Cell type markers were identified by differential expression analysis using the 'MAST' hurdle-model test (*Finak et al., 2015*). Genes overexpressed in a given cell type relative to all other cell types were considered cell type 'markers'. Cell type identity was assigned using previously reported markers for this data set (*Figure 5—figure supplement 1*).

Cells along the adipogenic trajectory were subset and subjected to dimensionality reduction (UMAP, dims = 1:10, resolution = 0.17). A range of resolutions from 0.01 to 0.25 were tested. Using *Adipoq* as a marker of differentiation, we sought to identify the set of clusters that would best encapsulate the stages of differentiation. To this end for every level of resolution we calculated the mean count variance ($\bar{C}_\sigma$). This is done by calculating the standard deviation ($\sigma$) of *Adipoq* expression (E) within each cluster (G), referred to as the count variance ($C_\sigma$). Cells with no expression of *Adipoq* were excluded. The mean of count variances for all clusters is calculated. This process is similar to k-means clustering, where the goal is to find that parameters which minimize the within group variation.

$$Count\ Variance = \frac{\sum_{G=1}^{n} \sigma(E_G)}{n}$$

We also calculated the percent expressing variance ($\bar{P}_\sigma$). This was taken as the mean of the standard deviation in the percent of cells expressing *Adipoq*.

$$Percent\ Expressing\ Variance = \frac{\sum_{G=1}^{n} \sigma(\%E>0_G)}{n}$$

The resolution 0.17 was chosen as the lowest resolution where variation is minimized and no longer significantly changes (*Figure 5—figure supplement 5*). Using *Adipoq* as a marker of adipogenesis, clusters 1 and 2 were identified as pre- and post-differentiated cells, respectively. Differential expression was analyzed using the 'MAST' test. Expression was compared between clusters 1 and 2 only. Multiple tests correction was performed using the Bonferroni method. We required changes in expression to show either a sufficiently large fold change ($|\log_2 FoldChange| \geq 0.3$) *OR* a sufficiently large change in the percent of cells expressing the gene in question ($pct.\Delta \geq 0.4$). The change in percent of cells expressing a gene was calculated as the difference in percent of cells expressing the gene between the clusters and scaled by dividing by the larger percentage.

$$pct.\Delta = \frac{pct.2 - pct.1}{\max(pct.1, pct.2)}$$

Source code is available at https://github.com/LawsonLab-WUSM/POE_Epistasis, (copy archived at swh:1:rev:b39046ce35f53e0c3f15bcdefa122c274aee48b7, *Lawson, 2019*).

## Acknowledgements

This work was supported by the Washington University Department of Genetics, the Diabetes Research Center at Washington University (grant P30DK020579), the NIH NIDDK (grant K01 DK095003), the NIEHS (grant U24ES026699), and the NIH NHGRI (grant T32-GM007067). The authors declare no conflicts of interest.

## Additional information

### Funding

| Funder | Grant reference number | Author |
|---|---|---|
| Diabetes Research Center | | Clay F Semenkovich |
| Diabetes Research Center | | Heather A Lawson |

The funders had no role in study design, data collection and interpretation, or the decision to submit the work for publication.

### Author contributions

Juan F Macias-Velasco, Formal analysis, Methodology, Visualization, Writing - original draft, Writing – review and editing; Celine L St Pierre, Visualization, Writing – review and editing; Jessica P Wayhart, Li Yin, Larry Spears, Caryn Carson, Katsuhiko Funai, Investigation; Mario A Miranda, Validation; James M Cheverud, Clay F Semenkovich, Resources, Writing – review and editing; Heather A Lawson,

Conceptualization, Funding acquisition, Methodology, Supervision, Writing - original draft, Writing – review and editing

### Author ORCIDs
Juan F Macias-Velasco http://orcid.org/0000-0003-2827-4647
Celine L St Pierre http://orcid.org/0000-0001-5465-6601
Clay F Semenkovich http://orcid.org/0000-0003-1163-1871
Heather A Lawson http://orcid.org/0000-0002-3550-5485

### Ethics

Mouse colony was maintained at the Washington University School of Medicine and all experiments were approved by the Institutional Animal Care and Use Committee in accordance with the National Institutes of Health guidelines for the care and use of laboratory animals. Protocol #20-0384.

### Decision letter and Author response
Decision letter https://doi.org/10.7554/eLife.72989.sa1
Author response https://doi.org/10.7554/eLife.72989.sa2

---

## Additional files

### Supplementary files
• Supplementary file 1. Allele-specific expression.

• Supplementary file 2. Biallelic genes differentially expressed by cross.

• Supplementary file 3. Networks of genes showing parent-of-origin allele-specific expression interacting with biallelic genes that are differentially expressed by cross.

• Supplementary file 4. Over-representation input/output.

• Supplementary file 5. $F_1$ Phenotype correlation.

• Supplementary file 6. Epistasis results.

• Supplementary file 7. Cell cluster markers.

• Supplementary file 8. Alignment summaries.

• Supplementary file 9. List of imprinted genes queried.

• Supplementary file 10. Genome annotations.

• Transparent reporting form

### Data availability

Sequencing data are available through the NCBI-SRA under accession code PRJNA753198.

The following dataset was generated:

| Author(s) | Year | Dataset title | Dataset URL | Database and Identifier |
|---|---|---|---|---|
| Lawson HA | 2021 | LG/J and SM/J hybrid RNAseq | https://www.ncbi.nlm.nih.gov/bioproject/PRJNA753198/ | NCBI BioProject, PRJNA753198 |

The following previously published dataset was used:

| Author(s) | Year | Dataset title | Dataset URL | Database and Identifier |
|---|---|---|---|---|
| Burl RB | 2018 | Single-cell RNA-sequencing of white adipose tissue stromal cells during CL-induced adipogenesis | https://www.ebi.ac.uk/ena/browser/view/PRJNA470640?show=reads | European Nucleotide Archive, PRJNA470640 |

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
