## [Editor Report]

We all learned simple Mendelian Punnett Squares in Junior High or earlier when studying simple Mendelian traits. But we also all know that the world is so much richer and more complex than this. The current article explores some of that complexity, opening rich insights into intergenerational effects, offering the opportunity for mathematical thinking and further hypothesis testing, and opening up exciting new hypotheses to test. As Professor Stephen Stearns wrote, "Many of us do not do science only, or even primarily, to achieve practical results. We do it because we are fascinated with neat ideas. Evolutionary medicine is full of them, including parent-of-origin pattern." Let us enjoy the wonder.

---

## [Decision Letter]

**Decision letter after peer review:**

Thank you for submitting your article "Parent-of-origin effects propagate through networks to shape metabolic traits" for consideration by *eLife*. Your article has been reviewed by 3 peer reviewers, and the evaluation has been overseen by a Reviewing Editor and Matthias Barton as the Senior Editor. The following individuals involved in review of your submission have agreed to reveal their identity: David Haig (Reviewer #1).

Essential revisions:

Overall, the reviewers are first and foremost requesting substantial modifications to clarify what exactly was done. Methodologic transparency will allow both the reviewers and the eventual readers to better understand what was done and its implications. Please respond to all of the reviewers' requests for more information and/or clarifications.

*Reviewer #1 (Recommendations for the authors):*

The model in which some parent-of-origin effects in crosses are mediated by effects of imprinted genes on the expression of unimprinted genes is probably the 'null hypothesis' for most workers in the field but it is good to have it spelled out.

Some evolutionary implications are briefly considered in the Discussion. The authors suggest that the various effects of Nnat in white adipose tissue, pancreatic β-cells, and bone are jointly critical for growth, consistent with the 'sexual conflict hypothesis' for the evolution of genomic imprinting.

I will make a minor technical comment on the authors' presentation of this hypothesis before considering the possible involvement of Nnat in thermogenesis. The hypothesis is presented as a conflict between the fitness of mothers and fathers. This is not quite correct, rather the relevant conflict arises from different 'inclusive' fitness’s of genes derived from mothers and genes derived from fathers (see Proc. R. Soc B 264: 1657). These two formulations are not the same because an offspring inherits only one of the two alleles present in a parent. Matrilineal and patrilineal inclusive fitness’s can differ when gene expression in one individual (the focal individual) has fitness consequences for other individuals with different probabilities of carrying the focal individual's maternal and paternal allele (call these other individuals 'asymmetric kin'). The category of asymmetric kin includes the mother of the focal individual, and this is where the conflict overgrowth discussed by the authors arises, but it also includes half-sibs, cousins, etc.

Thermogenesis and growth are, of course, not independent processes: fuels consumed to keep warm are unavailable for anabolic processes. Imprinting evolves for traits that mediate trade-offs between the fitness’s of self and others. Metabolism is likely to be central to these trade-offs.

"Imprinting-by-imprinting epistasis" is used in the main text but not defined. Its meaning is tucked away in the Methods section

"Nnat and F2r are negatively correlated along an adipogenic trajectory". "Nnat increases and F2r decreases along an adipogenic trajectory" would be more informative.

*Reviewer #3 (Recommendations for the authors):*

1) P-values are repeated in the text and in figures. Please see Figure 4C and D as an example.

2) Figures are out of order. For example, figure 5 I is referenced in the text before Figure 5 E-H.

3) In figure 4, the authors state "However, their interaction significantly correlates with basal glucose in the F2's (p=0.032)." This information seems like it should be in the text rather than in the figure legend.

4) In the text Figure 4F should be 4G and Figure 4G should be 4F.

5) It would be helpful for the reader to provide the direction of correlations in the text. For example, instead of stating "F2r expression significantly correlates with basal glucose…" include the word negatively or positively before the word "correlates."

6) Figure 5H was not referenced in the text.

7) Please include the N for group housing the mice.

8) The interpretation of the results seems a bit underdeveloped. The authors did all this great work, and it should be highlighted at the beginning of the discussion and then expanded upon. Rather, they lead with the second paragraph, which is unclear and appears to be out of place. What message are the authors trying to convey? Also, please include more references to support the statements being made and provide more context for the reader. For example, why would you expect F2r to be more strongly associated with glucose? Also, statements, such as "play some role…" are vague.

9) In "differential expression" section in the methods, please define TMM.

---

## [Author Response]

Reviewer #1 (Recommendations for the authors):The model in which some parent-of-origin effects in crosses are mediated by effects of imprinted genes on the expression of unimprinted genes is probably the 'null hypothesis' for most workers in the field but it is good to have it spelled out.Some evolutionary implications are briefly considered in the Discussion. The authors suggest that the various effects of Nnat in white adipose tissue, pancreatic β-cells, and bone are jointly critical for growth, consistent with the 'sexual conflict hypothesis' for the evolution of genomic imprinting.I will make a minor technical comment on the authors' presentation of this hypothesis before considering the possible involvement of Nnat in thermogenesis. The hypothesis is presented as a conflict between the fitness of mothers and fathers. This is not quite correct, rather the relevant conflict arises from different 'inclusive' fitness’s of genes derived from mothers and genes derived from fathers (see Proc. R. Soc B 264: 1657). These two formulations are not the same because an offspring inherits only one of the two alleles present in a parent. Matrilineal and patrilineal inclusive fitness’s can differ when gene expression in one individual (the focal individual) has fitness consequences for other individuals with different probabilities of carrying the focal individual's maternal and paternal allele (call these other individuals 'asymmetric kin'). The category of asymmetric kin includes the mother of the focal individual, and this is where the conflict overgrowth discussed by the authors arises, but it also includes half-sibs, cousins, etc.

Thank you for the comment. We have added text to clarify that the size of progeny in placental mammals can have opposing fitness consequences for mothers/ maternal relatives and fathers/ paternal relatives. We have also added references to the point that imprinting could have evolved in part to allow one parent to hijack parts of a nutritional environmental response pathway driving growth in a direction favorable to maximize the fitness of individuals sharing a parental line.

Thermogenesis and growth are, of course, not independent processes: fuels consumed to keep warm are unavailable for anabolic processes. Imprinting evolves for traits that mediate trade-offs between the fitness’s of self and others. Metabolism is likely to be central to these trade-offs.

This is a great point that we had not considered! We added language to reflect this as well as additional references.

"Imprinting-by-imprinting epistasis" is used in the main text but not defined. Its meaning is tucked away in the Methods section

We have defined this term when it is used in the main text.

"Nnat and F2r are negatively correlated along an adipogenic trajectory". "Nnat increases and F2r decreases along an adipogenic trajectory" would be more informative.

We have modified the language for clarity.

Reviewer #3 (Recommendations for the authors):1) P-values are repeated in the text and in figures. Please see Figure 4C and D as an example.

We removed the repeated p-values from the text where they are stated in the figures.

2) Figures are out of order. For example, figure 5 I is referenced in the text before Figure 5 E-H.

We have corrected this oversight.

3) In figure 4, the authors state "However, their interaction significantly correlates with basal glucose in the F2's (p=0.032)." This information seems like it should be in the text rather than in the figure legend.

We moved this statement to the text.

4) In the text Figure 4F should be 4G and Figure 4G should be 4F.

We have corrected this oversight.

5) It would be helpful for the reader to provide the direction of correlations in the text. For example, instead of stating "F2r expression significantly correlates with basal glucose…" include the word negatively or positively before the word "correlates."

We have modified the text as suggested for clarity.

6) Figure 5H was not referenced in the text.

We have corrected this oversight.

7) Please include the N for group housing the mice.

We have included this detailed information in the methods.

8) The interpretation of the results seems a bit underdeveloped. The authors did all this great work, and it should be highlighted at the beginning of the discussion and then expanded upon. Rather, they lead with the second paragraph, which is unclear and appears to be out of place. What message are the authors trying to convey? Also, please include more references to support the statements being made and provide more context for the reader. For example, why would you expect F2r to be more strongly associated with glucose? Also, statements, such as "play some role…" are vague.

We thank the reviewer for this comment. We have modified the discussion to expand upon our interpretation of the results, and included additional references.

9) In "differential expression" section in the methods, please define TMM.

We defined this method of normalization in the methods.